# *Coxiella burnetii* and Co-Infections with Other Major Pathogens Causing Abortion in Small Ruminant Flocks in the Iberian Peninsula

**DOI:** 10.3390/ani12243454

**Published:** 2022-12-07

**Authors:** María de los Angeles Ramo, Alfredo A. Benito, Joaquín Quílez, Luis V. Monteagudo, Cristina Baselga, María Teresa Tejedor

**Affiliations:** 1Department of Animal Pathology, Faculty of Veterinary Sciences, University of Zaragoza, Miguel Servet 177, 50013 Zaragoza, Spain; 2EXOPOL S.L, Pol Rio Gállego D/14, San Mateo del Gállego, 50840 Zaragoza, Spain; 3Agrifood Institute of Aragón (IA2), University of Zaragoza-CITA, Miguel Servet 177, 50013 Zaragoza, Spain; 4Department of Anatomy, Embryology and Genetics, Faculty of Veterinary Sciences, University of Zaragoza, Miguel Servet 177, 50013 Zaragoza, Spain; 5Centro de Investigación Biomédica en Red de Enfermedades Cardiovasculares CIBERCV, Faculty of Veterinary Sciences, University of Zaragoza, Miguel Servet 177, 50013 Zaragoza, Spain

**Keywords:** *Coxiella burnetii*, small ruminants, co-infections, abortion, Iberian Peninsula, vaccine

## Abstract

**Simple Summary:**

Abortions have a large economic impact in small ruminant flocks. Some abortions may be induced by non-infectious causes but most of them are due to a broad spectrum of abortifacient infectious agents. In this study, abortion cases from sheep (n: 1242) and goat (n: 371) flocks in the Iberian Peninsula were analyzed for the presence of *Coxiella burnetii* and other major abortifacients using molecular methods. *C. burnetii* and *Chlamydia abortus* were by far the most common pathogens circulating in sheep and goat flocks, since approximately 75% of cases from both hosts tested positive. *C. burnetii* infection was more prevalent in goats than sheep and was detected in almost half of caprine abortions. A significant proportion of *C. burnetii* PCR-positive cases were positive only for this bacterium in both ovine (33.1%) and caprine (63.2%) abortions, with the remaining cases harboring mixed infections. *Toxoplasma gondii* was the third most prevalent pathogen in both caprine (4.6%) and ovine (10.1%) abortions, followed by *Campylobacter* sp., *Salmonella enterica,* border disease virus and *Neospora caninum*. To the best of our knowledge, this is one of the most thorough studies reported in the literature regarding the presence of infectious agents in ovine and caprine abortions.

**Abstract:**

*Coxiella burnetii* is an intracellular bacterium causing human Q fever and reproductive disorders in domestic ruminants. We analyzed the occurrence of *C. burnetii* and co-infections with six other major pathogens causing abortion in sheep (1242 cases) and goat (371 cases) flocks from Spain and Portugal. After real-time PCR detection, co-infections were established by principal component and cluster analysis that grouped cases based on the joint presence/absence of several microorganisms. *C. burnetii* and *Chlamydia abortus* were the most common abortifacient agents with approximately 75% of cases from both hosts testing positive, followed by *Toxoplasma gondii*, *Campylobacter* sp., *Salmonella enterica,* border disease virus and *Neospora caninum*. *C. burnetii* was significantly more common than *C. abortus* in goat abortions (*p* < 0.001). Co-infections with at least two pathogens were found in more than 66% cases of ovine abortions and 36% cases of caprine abortions testing positive for *C. burnetii*, mostly including mixed infections with only *C. abortus.* These findings indicate that both pathogens are the most significant ones to be readily prevented by vaccination in this geographical area. Biosecurity and biocontainment measures are also steadfastly recommended to prevent both the economic losses and public health risks associated with most of these abortifacient agents.

## 1. Introduction

Abortions have a large economic impact in small ruminant livestock. Causes of abortion can be categorized as infectious and non-infectious, with the former being responsible for up to 90% of economic losses [1]. At the beginning of this century, annual losses due to abortions in sheep in the United Kingdom were estimated to be £20 million for chlamydial abortion and £12 million for toxoplasmosis [2]. More recently, direct economic losses due to outbreaks of toxoplasmosis in dairy and meat sheep flocks in Spain were estimated to be €171.8/abortion and €63.6/abortion, respectively [3]. The most common infectious agents causing abortions in sheep and goats are *Coxiella burnetii, Chlamydia abortus*, *Brucella* spp., *Leptospira* spp., *Campylobacter fetus*, *Salmonella enterica, Listeria* spp. *and Toxoplasma gondii*, with most of them being of zoonotic nature and representing a serious risk for human health [4,5].

*Coxiella burnetii* is an obligate Gram-negative intracellular bacterium infecting livestock, other domestic and wild mammals, birds and reptiles [6]. *C. burnetii* is also the causative agent of the zoonoses Q fever in humans, which is distributed worldwide and considered an emerging infectious disease in Europe [7,8]. Domestic ruminants (primarily sheep, goats and cattle) represent the most frequent source of human infection, which is mainly transmitted via inhalation of contaminated dust and aerosols [9]. *C. burnetii* shows a tropism for the trophoblasts of the placenta and the mammary gland, and is primarily shed via vaginal mucus, milk, birth products and faeces [10,11,12]. The impact of this infectious disease on human health was highlighted by the large Q fever outbreak affecting about 4000 people in the Netherlands during 2007–2010, causing economic losses of approximately €307 million, with aborting dairy goats being considered as the source [13,14].

In the European Union, there is no harmonized monitoring system in place for Q fever in animals, and the results of diagnosis from livestock are insufficient for tracking or analyzing trends [15,16]. *C. burnetti* infection in sheep and goats is mainly subclinical but can cause reproductive disorders in pregnant females, with abortions, stillbirth and delivery of weak offspring being the most common clinical signs of the disease [17]. Abortion rates of 10–60% have being suggested in sheep, and up to 72–81% have being reported for Q fever outbreaks in goat farms [18,19,20]. *C. burnetii* was determined as the most common cause of abortion in goats in the Netherlands, followed by *Chlamydia abortus* [21] but it was rarely seen in sheep in the United Kingdom, despite evidence of seropositivity [22,23]. Serological surveys have concluded that *C. burnetii* infection is widespread in livestock in many European countries, although seroprevalence is not correlated with bacterial shedding [24,25,26,27,28,29]. Molecular studies have also found that *C. burnetii* is a prevalent pathogen in sheep and goat abortion submissions, followed by *C. abortus*, which is a common concurrent pathogen [30]. Routine hygiene and cleaning measures are effective prevention tools. Vaccination, using a phase I *C. burnetii* inactivated vaccine, has also been reported to protect goats against abortion and reduce massive bacterial shedding from heavily infected herds [7,19].

In Spain, Q fever is considered an endemic nationally notifiable disease in humans [31,32]. The number of notified human Q fever cases has increased since 2013, which is mostly explained by the reporting system changing from voluntary to compulsory [33]. In the EU, Spain has reported the highest number of human cases annually since 2017, and has accounted for more than a third of the overall number of cases in 2019, which explains why human Q fever in this country is well documented in the literature [34]. In contrast, studies on the occurrence of *C. burnetii* infection in small ruminants are limited in Spain, and their impact in sheep and goat abortions remains unclear. Seroprevalence studies in semi-extensive grazing systems have shown herd seroprevalence values of up to 74% for sheep and 45% for goats, with individual seroprevalence values of 8.4% in sheep and 24.4–42% in goats [35,36,37]. Likewise, *C. burnetii* seems to be a widespread pathogen in environmental samples collected in Spanish livestock farms, as documented by the high percentage of dust and/or aerosols samples (36–80%) testing positive by real-time PCR analyses [37,38]. In this study, the current status of *C. burnetii* infection and other major pathogens, causing abortion in sheep and goat flocks from Spain and Portugal, was investigated through cases submitted for diagnosis to a veterinary laboratory. A statistical analysis of co-infections between *C. burnetii* and other pathogens was also conducted in order to determine the most significant microorganisms to be the target of vaccination to prevent abortions in these small ruminant hosts.

## 2. Materials and Methods

Samples were collected from ovine and caprine abortions by veterinary practitioners for diagnostic purposes after farm owners solicited their professional services to treat abortions in the herds, with no specific permits being required by the authorities for specimen collection. Committee approval for animal care and use was not needed for this study as stated by the Ethical Advisory Commission for Animal Experimentation of the University of Zaragoza (Ref. PI21/22NE). Directive 2010/63/EU of the European Parliament on the protection of animals used for scientific purposes does not apply to non-experimental clinical veterinary practices [39].

Samples from sheep and goat herds with a history of abortions were used. Specimens consisted of placenta, foetal organs, and/or vaginal/endocervical secretions (sterile swabs) from ovine and caprine abortions, including stillbirths, and were submitted to a veterinary diagnostic laboratory (Exopol SL, San Mateo de Gállego, Spain) from January 2017 to February 2022. Overall, samples from abortion episodes in a total of 1242 ovine and 371 caprine cases were analyzed. Most cases were submitted from sheep and goat flocks located in 43/50 provinces throughout Spain, although some cases originated from flocks in Portugal (Table 1). Samples were immediately refrigerated after collection and handed to the laboratory via urgent transport in less than 24 h. Vaginal swabs were obtained within the 48 h following abortion and also urgently transported. No data about vaccine programs against abortifacient infectious agents in these flocks were available.

Nucleic acid isolation was performed with the commercial kit MagMAX™ Pathogen RNA/DNA (Thermo Fisher Scientific, Waltham, MA, USA) and an automated magnetic particle processor (KingFisher Flex; Thermo Fisher Scientific, Waltham, MA, USA), according to the manufacturer’s instructions. Specimens were analyzed for detection of *Chlamydia abortus*, *Coxiella burnetii*, *Campylobacter* sp., *Salmonella enterica*, *Toxoplasma gondii*, *Neospora caninum* and border disease virus (BDV) by using real-time PCR (qPCR) and reverse transcription real-time PCR (RT-qPCR) commercial kits (EXOPOL S.L.U., San Mateo de Gállego, Spain), according to the manufacturer’s protocol and instructions. These assays target the ompA gene, IS111 sequence, gyrB gene, ttrC gene, repeat region B1, NC5 gene and 5´UTR region respectively. Samples were considered positive based on a cycle threshold (Ct) value of <38. It should be noted that not all of these target pathogens were analyzed for some abortion cases. The prevalence of *Brucella* sp. has declined in response to official eradication campaigns in Spain and was not analyzed in this study; in fact, the European Commission has officially declared Spain to be free of ovine and caprine brucellosis.

The IBM SPSS Statistics 26.0 software package (IBM Corp., Armonk, NY, USA) was used for statistical analyses. Comparison of detection frequencies was carried out by chi-squared test. The study of co-infections acquired a statistical basis, thanks to the use of principal component analysis (PCA). PCA was applied to co-infection by two or more of these microorganisms: *C. abortus*, *C. burnetii*, *T. gondii*, *Campylobacter* sp., *S. enterica*, BDV and *N. caninum*. Only samples with valid data for all these pathogens were included in this analysis. A set of seven categorical variables relative to absence (0) or presence (1) of the considered pathogens was reduced into a smaller set of new variables (principal components); these new variables accounted for most of the variance in the original variables. Decision about the number of retained components was made on the basis of the eigenvalue-one criterion [40] and the scree plot [41]; orthogonal rotation (Varimax) was carried out.

A cluster analysis, applied to the retained principal components, allowed grouping cases based on the joint presence (or absence) of several microorganisms. Using a hierarchical agglomerative procedure, the number of clusters was defined by the elbow rule (Ward’s method). The k-means procedure was applied to actually form the clusters [42]. The resulting clusters were groups of cases that shared similar characteristics related to the presence or absence of the considered microorganism; on the basis of the observed frequencies and the significant associations of the studied microorganisms, these characteristics would point to the main co-infections of *C. burnetii* with other infectious agents.

## 3. Results

A summary of the number of cases analyzed and their distribution according to the country of origin, year and season is indicated in Table 1. Most abortion cases (>90%) from both sheep and goats originated in flocks in Spain with a few specimens being submitted from farms in Portugal (1.9–4.3%) and some cases of unknown origin. The number of cases analyzed every year was quite similar (15.1–23.5% per year), except for 2022 when only samples received in January and February were analyzed. The distribution of cases along the seasons was uneven, with >60% of specimens being submitted in autumn and winter.

The number of cases testing positive by molecular analyses for any of the evaluated pathogens is indicated in Table 2. The pathogens most commonly diagnosed in sheep and goat abortions were *C. burnetii* and *C. abortus*, with approximately 75% of cases from both small ruminant hosts testing positive for these pathogens. The protozoan *T. gondii* was the third most prevalent abortifacient agent identified in both sheep (10.1%) and goats (4.6%), followed by *Campylobacter* sp., *S. enterica*, BDV and *N. caninum.*

A total of 1150 samples from ovine abortions, possessing data for the seven major pathogens, were submitted for PCA and cluster analysis. Four principal components were retained; the cumulative percentage of variance they explained was 62.7%. Table 3 shows the main results from PCA, and the rotated component matrix. This matrix includes the correlation coefficients for each original variable and the estimated principal components. PCA simplified the information from the original seven variables by summarizing them in only four principal components (1, 2, 3 and 4), each of them including the original variables shown in Table 3. For clarity, coefficients below 0.3 (absolute value) were suppressed. Positive loading indicates a positive correlation between a variable and a principal component: hence, they move in the same direction. On the contrary, negative loadings mean a negative correlation; the variable and the principal component move in opposite directions. When a variable has a strong effect on a principal component, the loading value shows a large absolute value. Principal component 1 loads very strongly on *T. gondii and C. abortus* (negative load); this principal component mainly compiles the information about the presence of *T. gondii* along with the absence of *C. abortus*. Principal component 2 loads strongly on *C. burnetii* and BDV, while it shows a moderate negative load for *N. caninum*; hence, principal component 2 summarizes the information about the presence of both *C. burnetii* and BDV, and the absence of *N. caninum*. Principal component 3 loads very strongly on *S. enterica* and moderately on *N. caninum*; this principal component mainly includes the information about the presence of both agents. Finally, principal component 4 loads very strongly on *Campylobacter* sp.

Once PCA reduced the original variables set, the study of co-infections was carried out by means of cluster grouping, based on the four retained principal components. Therefore, the resulting clusters grouped cases showing similar characteristics to the variables included in the four principal components; a simplified view of the most relevant co-infections in the considered population was achieved. Table 4 shows the results of the cluster analysis applied on the four retained principal components in sheep abortions. Each cluster dealt with a particular pathogen, because most cases showing this microorganism were enclosed in it. Clusters 1, 2 and 3 included most cases positive for BDV (49/52; 94.2%), *S. enterica* (72/85; 84.7%) and *Campylobacter* sp. (75/84, 89.3%), respectively. Most abortions positive for *C. burnetii* (338/444, 76.1%) were included in Cluster 5, which also allocated most cases testing positive for *C. abortus* (397/467, 85.01%) and *T. gondii* (105/115, 91.30%). Cluster 7 involved most of the *N. caninum* positives (30/33; 90.9%). All nine abortion cases allocated to Cluster 4 tested positive for concurrent infections between *Campylobacter* sp. and up to three other pathogens, and mostly with *S. enterica* (7/9). Finally, all three abortion cases allocated in cluster 6 were positive for both *N. caninum* and *S. enterica*.

A more accurate description of co-infections between *C. burnetii* and other pathogens in ovine abortions and their distribution in the different clusters is indicated in Table 5. As a summary, co-infections with at least two pathogens were found in more than 66% cases testing positive for *C. burnetii* (297/444), mostly including mixed infections with only *C. abortus* (145/444, 32.6%). A total of 147 cases (33.1%) were positive for only *C. burnetii*.

A total of 335 samples from caprine abortions had data for the seven major pathogens and were submitted for PCA and cluster analysis. Three principal components were retained, accounting for 49.0% of total variance. Table 6 shows the results from PCA (rotated component matrix); only coefficients above 0.3 (absolute value) were included. Principal component 1 loads strongly on *C. abortus* (negative value) and moderately on *T. gondii* and *S. enterica*. Principal component 2 loads strongly on BDV and *Campylobacter* sp. Principal component 3 loads very strongly on *N. caninum* and strongly on *C. burnetii*.

Cluster grouping based on the three retained component analyses was applied to study co-infections with *C. burnetii* and the results are shown in Table 7. The two samples positive for *N. caninum* were included in cluster 1. Cluster 2 only included one sample positive for both *Campylobacter* sp. and BDV. Cluster 3 allocated the remaining three samples positive for BDV. Cluster 4 included all samples positive for *S. enterica* (6/6) and most (12/13) samples positive for *Campylobacter* sp. Most samples positive for *T. gondii* (13/16) were allocated to cluster 5. Finally, cluster 6 included most samples positive for *C. burnetii* (148/163, 90.8%) and *C. abortus* (84/88, 95.4%).

A detailed description of co-infections between *C. burnetii* and other pathogens in caprine abortions is shown in Table 8. In summary, co-infections with at least two pathogens were found in more than 36% cases testing positive for *C. burnetii* (60/163), mostly due to mixed infections with *C. abortus* (44/163, 27%). A total of 103 cases (63.2%) were positive for only *C. burnetii*.

## 4. Discussion

The rate of abortion in healthy herds is usually below 2%, but sheep and goat flocks can experience higher abortion rates sometimes resulting in “storms”, which significantly affect productivity and require accurate diagnosis [4]. Some abortions in the present report may be induced by other possible abortifacient pathogens not included in our panel or by non-infectious causes, which have been reported to account for approximately 10% of events [43,44]. Non-infectious causes were not investigated in this study. Among the 1613 specimens studied in this work, a remarkable percentage tested negative to all the seven pathogens analyzed, namely more than 23% of cases from sheep and 33% from goats, suggesting a potential non-infectious etiology, or the action of other infectious agents. Several factors have been documented to cause abortions in livestock, including toxic chemicals, genetic factors, metabolic and nutritional problems and physical factors. Non-infectious abortions are mostly sporadic and occur at any pregnancy stage, but causes are difficult to diagnose and in most cases remain unknown [43,44]. Nevertheless, available evidence indicates that different bacteria, viruses and protozoa are by far the most plausible causative agents of abortions in sheep and goats [45].

In this study, we investigated the occurrence of the most common infectious causes of abortion in specimens submitted by veterinarians to an animal health laboratory. A panel of commercial qPRC and RT-qPCR assays were used for a rapid diagnosis of these pathogens. These molecular methods have been reported to provide highly sensitive, specific and rapid tools for the specific detection of different abortifacient infectious agents in various clinical samples, with multi-screening qPCR approaches allowing the simultaneous detection of the main abortive agents [45,46,47]. It is generally accepted that both standard PCR and qPCR provides diagnostic information on pathogens existing in samples [48]. High qPCR values provide strong evidence that infectious abortions are caused by a specific pathogen and, therefore, play a key role in the management of abortions in sheep and goat herds; however, definitive diagnosis should also consider other indicators, such as histopathological lesions, microorganism cultures and specific staining techniques [28,30]. A cycle threshold (Ct) value of 38 was selected in the current study based on the limit of detection of the assay. It means that positive RT-qPCR results that are above this threshold have a likelihood of being a false positive, especially considering that different pathogens and specimens/tissues were analyzed. Another limitation of this study was related to the collection of specimens by veterinary clinics, and the risk of the contamination of samples in heavily contaminated environments, which could even provide low Ct values in RT-qPCR.

In the current study, *Coxiella burnetii* and/or *Chlamydia abortus* were identified in approximately 75% of ovine and caprine abortions. Both pathogens were found in a similar percentage of abortions in sheep but *C. burnetii* was significantly more common than *C. abortus* in goat abortions (*p* < 0.001). Additionally, a high proportion of *C. burnetii* PCR-positive cases were positive for only this bacterium in both ovine (33.1%) and especially caprine (64%) abortions. These findings support the potential role of the latter bacterium as a major pathogen in livestock abortion in this geographical area, with goat herds being more sensitive to *C. burnetii* infection, as previously reported [49]. Q fever is well recognized as a cause of economic losses in small ruminant farms through increased abortion rates and the loss of milk production [50]. Goats are considered to be more severely affected than sheep by *C. burnetii* infections, as they suffer more abortions and deliver more weak offspring, with some studies reporting abortion rates of up to 90% [51,52]. Abortion waves in dairy goats have also been considered the cause of large outbreaks of Q fever, affecting more than 4000 people in the Netherlands [53]. Nevertheless, few studies have investigated the occurrence of infectious agents in sheep and goat abortions in European farms and data are difficult to compare, because of the differences in the methodology or the composition of the populations studied [49].

A review on the prevalence of *C. burnetii* in small ruminants revealed that the majority of the published studies dealt with seroprevalence rather than shedding prevalence, and most of the latter were based on the testing of milk samples only [11]. *C. burnetii* was implicated as the primary cause of abortions in sheep in Hungary and was identified by PCR in a dairy goat herd from the UK reporting an outbreak with 22% abortions [54,55]. A high occurrence of *C. burnetii* has also been reported in Poland, with 51% of goat herds and 22% of sheep flocks testing positive by RT-qPCR [56]. In France, the proportion of abortive episodes potentially related to *C. burnetii* was 6.2% in sheep and 16.7% in goats [57]. In Greece, specimens from 11/65 sheep and goat farms experiencing abortion episodes tested positive for *C. burnetii* by qPCR [58]. In contrast, PCR results demonstrated that *C. burnetii* plays a relatively low role in abortion in Italy, especially in goats [24]. In Spain, the occurrence of this pathogen in small ruminant abortions is not well documented but Q fever outbreaks have been described in goat herds affecting up to 81% of females [20]. *C. burnetii* was detected in only 3% of sheep flocks, with abortion events in northern Spain using staining and serological methods [59]. Serological surveys have shown that a remarkable percentage of domestic ruminants have been in contact with *C. burnetii*, and goats and sheep have been recognized as risk factors in various sporadic and epidemic cases of human Q fever in Spain [35,37,60].

Concurrent infections with at least two pathogens were found in a remarkable number of abortion cases in both sheep (29.6%) and goats (18.5%) in this study; this revealed the challenge of unravelling the major contributing causes of abortion, because more than one agent is often involved in the same outbreak. *C. burnetii* plus *C. abortus* was the most common mixed infection, with both being the only pathogens reported in approximately one third of the cases testing positive for *C. burnetii*. Both bacteria have been reported to impact significantly upon Portuguese sheep and goat farms, where a significant proportion of abortion cases tested positive for *C. abortus* (34.2%), *C. burnetii* (15.1%) and co-infections between both pathogens (16.4%) [61]. In Italy, 11.3% of fetuses and 11.8% samples of ovine placenta yielded positive PCR results for two or more infectious agents [62]. In Ireland, 6% of farms with naturally occurring outbreaks of ovine abortion reported evidence of dual infection between *C. abortus* and *T. gondii* using a PCR assay [63].

*C. abortus* causes ovine and caprine enzootic abortion in sheep, which is also a major cause of lamb and goat kid loss in Europe [64]. This disease is by far the most commonly diagnosed cause of abortion in the UK, accounting for 42% of all cases during the four-month period of 2020 (177 out of 420 cases). The cost of enzootic abortion to the UK sheep industry was estimated to be up to £20 million [65]. *C. abortus* was detected in 50% of clinically healthy sheep flocks by PCR testing in Germany [66]. In Spain, published data on the detection of *C. abortus* in small ruminant abortions are very limited. Analysis by immunohistochemistry and PCR of samples of placenta from aborted sheep and goats in the south-east of the country revealed that it was the most prevalent abortifacient agent, although cross-sectional serological surveys have shown large variations in reported seroprevalence [67]. A study of sheep flocks in central Spain revealed that individual seroprevalence was 50.5% and the herd seroprevalence was 96.5% [68]. In contrast, seroprevalence of *C. abortus* was only 3.9% in semi-intensive lamb-producing flocks in northwest Spain, with a farm-level exposure of 18.2% [69]. A more recent study showed an overall seroprevalence of 33% in unvaccinated goat herds on the Canary Islands [70].

Two obligate intracellular protozoan parasites potentially causing abortion in livestock were also analyzed in this study. *T. gondii* is recognized as a serious cause of fetal mortality in sheep and goats and was the third most common abortifacient agent identified in this study, with 10.1% of sheep and 4.6% of goat submissions testing positive. In sheep, *T. gondii* was commonly accompanied by other pathogens, mostly *C. burnetii* and *C. abortus*, while most *T. gondii* positive cases in specimens from goats tested negative to other pathogens. In contrast, *N. caninum*, which is considered to be one of the most relevant abortifacients in dairy and beef cattle, was identified in less than 3% and 1% of specimens from sheep and goats, respectively, revealing its minor role as a cause of reproductive disorders in small ruminants. In Europe, there is little information on the occurrence of *T. gondii* as a cause of ovine and caprine abortion outbreaks. *T. gondii* is the second most common cause of abortions in sheep in the UK [65], and was the abortifacient most commonly detected in both ovine (18.1%) and caprine (13%) fetuses in Sardinia, Italy [62]. This protozoon was responsible for a massive outbreak of abortions (60%) in a Greek dairy sheep flock [71]. Rates of *T. gondii*-specific DNA detected in ovine-abortion derived tissues submitted for diagnosis were 10% in Ireland and 4.5% in Germany [63,72]. In Spain, studies with samples from farms with naturally occurring abortion outbreaks displayed *T. gondii* PCR-positive rates similar to those reported in the current study, ranging between 5.4% (4/74), 6.9% (12/173) and 16.9% (9/53) for ovine fetuses, and 3.8% (1/26) for caprine fetuses [73,74,75].

Studies on the prevalence of *N. caninum* in ovine and caprine flocks with confirmed reproductive failure are also limited in Europe. Although this protozoon has caused serious reproductive losses in some Spanish flocks, most of the available studies suggest that it is detected only sporadically and has, therefore, traditionally been considered as a minor parasite in small ruminants [76,77]. *N. caninum* was detected in 2.4% and 1.4% of the tested ovine and caprine fetuses in Switzerland, 3.5% of tissue samples from ovine abortions in Germany, and 8.6% of ovine fetuses in Italy [62,72,78]. In Spain, a previous survey with specimens submitted for diagnosis to a laboratory in 2008 and 2009 displayed higher rates than those detected in the current study, namely 6.8% of sheep fetuses (5/74) and 11.5% of goat fetuses (3/26) were PCR-positive for *N. caninum* [75].

Other infectious causes of abortion analyzed in this study (*Campylobacter* sp., *S. enterica* and Pestivirus causing border disease, BDV) each displayed <8% and <4% of PCR-positive ovine and caprine-abortion derived samples, respectively, suggesting that they are minor contributors to reproductive failure in the studied area. Infections with *Campylobacter jejuni*, *Campylobacter fetus* subsp. *fetus* and *S. enterica* (*S. enterica* ssp. *enterica* serovar *Abortusovis*) in pregnant ewes can cause abortion storms in late pregnancy or the full-term birth of dead or weak lambs. Infection with BDV in sheep can cause fetal death at any stage of pregnancy, but is more common in fetuses infected early in gestation. In accordance with the results of this study, abortions by all these pathogens can also occur in goats, but less commonly than in sheep [79,80,81,82]. Nevertheless, it is significant to mention that these agents can also be major abortifacients in sheep and goat flocks in other geographical areas. Campylobacteriosis ranks third after enzootic abortion and toxoplasmosis as a cause of abortion in sheep the UK [65], and was the most detected disease (19%) in ovine abortions during the 2012–2013 lambing season in the Netherlands [83]. The abortion rate caused by *S. enterica* is often in the range of 30–50%, based on outbreaks in endemic regions, but rates up to 90% have been reported [82]. *S. enterica* was identified in five sheep flocks with a high rate of abortion (22–38%) during the last-third of gestation in Croatia [84]. BDV infection is also globally distributed and has been reported in different European countries [85]. In northern Spain, an epidemic outbreak with an unusually high mortality caused by BVD was described in 1997 [86]. Subsequent studies in the same geographical area revealed that border disease and toxoplasmosis rank first or second as the most common infectious causes of abortion in sheep, with salmonellosis being reported in 7–10% of the flocks [59,87]. Comparison of these data with findings from our study indicates that the ranking of infectious causes of abortion in small ruminants is not fixed and can change depending on the study period or geographical area.

Management of abortions in livestock may be achieved through a combination of biosecurity and biocontainment measures, although vaccination is one of the most effective tools. Measures based on reducing or preventing the introduction of new diseases onto the farm from outside sources should be implemented. In addition to routine hygiene and disinfection of housing facilities, aborted ewes and those giving birth to dead full-term lambs should be isolated immediately, and aborted material and infected bedding removed and destroyed [4]. In Europe, vaccination against enzootic abortion has been achieved using inactivated whole organism-based vaccines or live attenuated vaccines, based on the 1B strain of *C. abortus* (Cevac Chlamydia^®^, Ceva Animal Health, Barcelona, Spain; Enzovax^®^, MSD Animal Health, Salamanca, Spain; Inmeva^®^; Laboratorios Hipra S.A., Amer, Spain) [88]. An inactivated vaccine, combining *S. enterica* and *C. abortus* (Inmeva^®^ Laboratorios Hipra S.A., Amer, Spain), has been shown to provide a 75% reduction in enzootic abortion rate and 55% reduction in the shedding of *C. abortus* [89]. Vaccination against *C. burnetii* can be achieved using an inactivated phase I vaccine (Coxevac^®^, Ceva Animal Health, Barcelona, Spain), which is approved for the active immunization of cattle and goats. Sheep are not a target species for Coxevac^®^, although field studies have shown that vaccination is effective in reducing massive bacterial loads in the environment and reducing the risk of future Q fever outbreaks in uninfected animals [90,91]. In contrast, the phase II *C. burnetii* vaccine did not affect the course of the disease or bacterial excretion [92]. The only authorized vaccine against *T. gondii* in sheep is Toxovax^®^ (MSD Animal Health, Salamanca, Spain), which is a live vaccine used to reduce tissue cyst development of the protozoan [93] Currently, there are no proven effective vaccines for border disease and some sheep farmers use vaccines directed against the related Bovine Viral Diarrhea Virus (BVDV); however, recent studies have shown that they do not protect against fetal infection after the challenge of treating pregnant ewes with BDV [94].

## 5. Conclusions

To the best of our knowledge, this is one of the most extensive studies reported in the literature on the presence of abortifacient infectious agents in ovine and caprine specimens, considering both the number of samples tested and the time over which specimens were available, even if diagnostics are limited to qPCR procedures and no data on histology or on antibody titration are available. *C. burnetii* and *C. abortus* were by far the most prevalent pathogens circulating in sheep and goat flocks and, therefore, should be the main targets of vaccination to prevent abortions in small ruminants in this geographical area. Further studies are required to investigate and approve the vaccine against *C. burnetii* in sheep, in response to the significant role played by this bacterium. Nevertheless, other pathogens were also identified in a not insignificant number of abortions, especially in sheep and frequently from mixed infections with *C. burnetii* and *C. abortus*, which advise vaccination against other bacteria (*Campylobacter* sp., *S. enterica*) and protozoa (*T. gondii*) whenever possible. Moreover, considering the geographical variations in the ranking of infectious causes of abortion reported in other studies, it is recommended that investigations are conducted to establish the most common abortifacient agents to be prevented by vaccination at a local or regional level. Biosecurity and biocontainment measures are also critical to prevent both economic losses and public health risks associated with most of these infectious agents.

## Figures and Tables

**Table 1 animals-12-03454-t001:** Distribution of the ovine and caprine abortion cases studied according to the country of origin, year and season. Data are counts (%). ND: not determined.

		Sheep (n = 1242)	Goat (n = 371)
	Spain	1130 (91.0%)	336 (90.6%)
**Origin**	Portugal	24 (1.9%)	16 (4.3%)
	ND	88 (7.1%)	19 (5.1%)
**Year**	2017	203 (16.3%)	77 (20.8%)
	2018	230 (18.5%)	56 (15.1%)
	2019	235 (18.9%)	65 (17.5%)
	2020	272 (21.9%)	72 (19.4%)
	2021	260 (20.9%)	87 (23.5%)
	2022	42 (3.4%)	14 (3.8%)
**Season**	Winter	484 (39.0%)	138 (37.2%)
	Spring	256 (20.6%)	64 (17.3%)
	Summer	217 (17.5%)	69 (18.6%)
	Autumn	285 (22.9%)	100 (27.0%)

**Table 2 animals-12-03454-t002:** Pathogens identified by qPCR and RT-qPCR in the ovine and caprine abortion samples studied from January to February 2022. Data are positive count/n, (%), 95% confidence interval limits [LL: low limit, UL: upper limit].

Pathogen	Sheep	Goats
*Coxiella burnetii*	483/1241 (38.9%) [33.8%, 44.0%]	180/371 (48.5%) [43.4%, 53.6%]
*Chlamydia abortus*	492/1210 (40.7%) [37.9%, 43.5%]	91/348 (26.1%) [21.5%, 30.7%]
*Toxoplama gondii*	119/1176 (10.1%) [9.2%, 11.0%]	16/345 (4.6%) [3.5%, 5.7%]
*Campylobacter* sp.	89/1162 (7.7%) [6.2%, 9.2%]	13/335 (3.9%) [2.9%, 4.9%]
*Salmonella enterica*	87/1165 (7.5%) [6.0%, 9.0%]	6/337 (1.8%) [0.4%, 3.2%]
border disease virus	52/1165 (4.5%) [3.3%, 5.7%]	4/337 (1.2%) [0.0%, 2.4%]
*Neospora caninum*	33/1157 (2.9%) [1.9%, 3.9%]	2/335 (0.6%) [0.0%, 1.4%]

**Table 3 animals-12-03454-t003:** Rotated component matrix of pathogens identified in ovine abortions: loadings of principal components 1, 2, 3 and 4 on variables related to absence/presence of pathogens in ovine samples. Coefficients below 0.3 (absolute value) have been suppressed for easier interpretation.

Variable	Principal Component			
	1	2	3	4
*Toxoplasma gondii*	0.807			
*Chlamydia abortus*	−0.722		−0.334	
*Coxiella burnetii*		0.679		
border disease virus		0.607		−0.345
*Salmonella enterica*			0.729	
*Neospora caninum*		−0.463	0.499	
*Campylobacter* sp.				0.893

**Table 4 animals-12-03454-t004:** Characteristics of samples from 1150 ovine abortions included in the seven clusters obtained in the study.

Cluster	Frequency	*C. burnetii*		*Campylobacter* sp.		*C. abortus*		*N. caninum*		Border Disease Virus		*S. enterica*		*T. gondii*	
		−	+	−	+	−	+	−	+	−	+	−	+	−	+
1	49 (4.27%)	26	23	49		30	19	49			49	46	3	48	1
2	72 (6.26%)	39	33	72		49	23	72		72			72	68	4
3	75 (6.52%)	40	35		75	54	21	75		75		75		73	2
4	9 (0.78%)	3	6		9	7	2	9		6	3	2	7	8	1
5	912 (79.3%)	574	338	912		515	397	912		912		912		807	105
6	3(0.26%)	3		3		3			3	3			3	3	
7	30 (2.61%)	21	9	30		25	5		30	30		30		28	2
Total	1150	706	444	1066	84	683	467	1117	33	1098	52	1065	85	1035	115

Absence or presence of a particular pathogen is recorded as − and +, respectively.

**Table 5 animals-12-03454-t005:** Number of ovine abortion cases positive for *C. burnetti*, and co-infections between *C. burnetii* and other pathogens in seven clusters indicated in Table 4. Data are count (%).

Cluster	*C. burnetii* Positive Cases (n = 444)	Co-Infection Frequency	Co-Infection Type
1	23	11 (2.5%)	border bisease birus
		10 (2.2%)	border disease virus, *C. abortus*
		1 (0.2%)	border disease virus, *T. gondii*
		1 (0.2%)	border disease virus, *S. enterica*
2	33	18 (4.0%)	*S. enterica*
		13 (2.9%)	*S. enterica*, *C. abortus*
		1 (0.2%)	*S. enterica*, *T. gondii*
		1 (0.2%)	*S. enterica*, *C. abortus*, *T. gondii*
3	35	23 (5.2%)	*Campylobacter* sp.
		10 (2.2%)	*Campylobacter* sp., *C. abortus*
		2 (0.4%)	*Campylobacter* sp., *T. gondii*
4	6	2 (0.4%)	*S. enterica*, *Campylobacter* sp.
		2 (0.4%)	*Campylobacter* sp., Pestivirus
		1 (0.2%)	*S. enterica*, *Campylobacter* sp., *C. abortus*
		1 (0.2%)	*S. enterica*, *Campylobacter* sp., *C. abortus*, *T. gondii*, *Pestivirus*
5	338	147 (33.1%)	None
		145 (32.6%)	*C. abortus*
		36 (8.1%)	*T. gondii*
		10 (2.2%)	*C. abortus*, *T. gondii*
7	9	6 (1.3%)	*N caninum*
		2 (0.4%)	*N caninum*, *C. abortus*
		1 (0.2%)	*N caninum*, *T. gondii*

**Table 6 animals-12-03454-t006:** Rotated component matrix of pathogens identified in caprine abortions: loadings of principal components 1, 2 and 3 on variables related to absence/presence of pathogens in caprine samples. Coefficients below 0.3 (absolute value) have been suppressed for easier interpretation.

Variable	Principal Component		
	1	2	3
*Chlamydia abortus*	−0.673		
*Toxoplasma gondii*	0.513		
*Salmonella enterica*	0.42		
border disease virus		0.711	
*Campylobacter* sp.	0.327	0.694	
*Neospora caninum*			0.801
*Coxiella burnetii*	−0.301		0.602

**Table 7 animals-12-03454-t007:** Characteristics of samples from 335 caprine abortions included in the six clusters obtained in the study. Absence or presence of a particular pathogen was recorded as 0 and 1, respectively.

Cluster	Frequency	*C. burnetti*		*Campylobacter* sp.		*C. abortus*		*N. caninum*		Border Disease Virus		*S. enterica*		*T.gondii*	
		−	+	−	+	−	+	−	+	−	+	−	+	−	+
1	2 (0.6%)		2	2		2			2	2		2		2	
2	1 (0.3%)	1			1	1		1			1	1		1	
3	3 (0.9%)		3	3		1	2	3			3	3		3	
4	17 (5.1%)	9	8	5	12	15	2	17		17		11	6	15	2
5	13 (3.9%)	11	2	13		13		13		13		13			13
6	299 (89.3%)	151	148	299		215	84	299		299		299		298	1
Total	335	172	163	322	13	247	88	333	2	331	4	329	6	319	16

**Table 8 animals-12-03454-t008:** Number of caprine abortion cases positive for *C. burnetti* and co-infections between *C. burnetii* and other pathogens in seven clusters indicated in Table 7. Data are positive count, (%).

Cluster	*C. burnetii* PositiveCases (n = 163)	Co-Infection Frequency	Co-Infection Type
1	2	2 (1.2%)	*N. caninum*
3	3	3 (1.8%)	border disease virus
4	8	4 (2.4%)	*Campylobacter* sp.
		1 (0.6%)	*Campylobacter* sp., *C. abortus*, *T. gondii.*
		1 (0.6%)	*Campylobacter* sp., *S. enterica*
		1 (0.6%)	*S. enterica*
		1 (0.6%)	*S. enterica*, *T. gondii*
5	2	2 (1.2%)	*T. gondii*
6	148	103 (63.2%)	None
		44 (27.00%)	*C. abortus*
		1 (0.6%)	*C. abortus*, *T. gondii*

## Data Availability

The data sets analyzed during the current study are available from the corresponding author on reasonable request.

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
