# Peer review of "Coxiella burnetii and Co-Infections with Other Major Pathogens Causing Abortion in Small Ruminant Flocks in the Iberian Peninsula"

_animals, 2022, doi:10.3390/ani12243454_

Round 1
Reviewer 1 Report
The manuscript reports the detection of pathogens related to abortion cases in small ruminants, which is very important. The sample size allows the data to be analyzed and statistically correlated.
In the discussion (line 276), the scope of PCR and qPCR methods in the detection of pathogens is mentioned. However, in this study, the records are only qualitative (presence or absence of genetic material of pathogens), so it is not relevant to add information on these differences. It is recommended to synthesize it or, alternatively, to discuss its relation with this analysis.
Punctuaction in the legend of Table 4 should be corrected.
Pathogens identified by qPCR and RT-qPCR in the ovine and caprine abortion samples 179 studied from January to February 2022" The sample sizes do not correspond to those presented in Table 1 and in the text.
Revise and homogenize the format of the bibliography.
Author Response
Reviewer 1
Comments and Suggestions for Authors
The manuscript reports the detection of pathogens related to abortion cases in small ruminants, which is very important. The sample size allows the data to be analyzed and statistically correlated.
In the discussion (line 276), the scope of PCR and qPCR methods in the detection of pathogens is mentioned. However, in this study, the records are only qualitative (presence or absence of genetic material of pathogens), so it is not relevant to add information on these differences. It is recommended to synthesize it or, alternatively, to discuss its relation with this analysis.
Thanks for the suggestion. This paragraph has been modified. The new text is:
It is generally accepted that both standard PCR and qPCR provides diagnostic information on pathogens existing in samples [48]. High qPCR values provide strong evidence that infectious abortions are caused by a specific pathogen and thus play a key role in the management of abortions in sheep and goat herds although definitive diagnosis should also consider other indicators such as histopathological lesions, microorganism cultures and specific staining techniques [28, 30]. A cycle threshold (Ct) value of 38 was selected in the current study based on the limit of detection of the assay. It means that positive RT-qPCR results that are above to this threshold have a likelihood of being false positive, especially considering that different pathogens and specimens/tissues were analyzed. Another limitation of the study was related with the collection of specimens by veterinary clinics and the risk of contamination of samples in heavily contaminated environment which could even provide low Ct values in RT-qPCR.
Punctuaction in the legend of Table 4 should be corrected.
Table 4 has suffered small modifications, in agreement with other referee. The format has been revised, and we think it is correct in its present form.
Pathogens identified by qPCR and RT-qPCR in the ovine and caprine abortion samples 179 studied from January to February 2022" The sample sizes do not correspond to those presented in Table 1 and in the text.
The global figures are not totally coincident because some pathogens were not searched in all the samples. In fact, no agents were analysed in the total set of 1242 samples. This fact is mentioned in the line 139: (“It should be noted that not all these target pathogens were analyzed for some abortion cases”)
Revise and homogenize the format of the bibliography.
Thanks again for you effort by supervising the references list. We have carefully revised it. Some references come from electronic revues, and the format of the reference does not contain volume, pages etc. Other references have no DOI (mainly the least modern ones). In all we think the current list is correctly written.

Reviewer 2 Report
Dear authors,
I enjoyed reading your manuscript. I do have some small suggestions. Two things that, to my opinion, should be discussed more extensively in the discussion are:
1. I do have some hesitations with the interpretation of Ct values of the Coxiella burnetii PCR. Low Ct values of this ubiquitous bacterium can also be caused by contamination (see Roest et al., 2012), especially in heavily contaminated environment this is a well know phenomenom. This ris for contamination is also the case for post mortem rooms. How sure are you that the samples with low Ct- values are not caused by contamination?
2. PCR detects a pathogen, but no causality with the abortion is shown since this manuscript does not contain pathological findings (and thus prove whether inflammatory signs fitting a certain pathogen is lacking)
Another point is the advise to vaccinate against several pathogens. To my opinion more debate on the advantages and disadvantages of some of the vaccines should be added. Vaccination against coxiellosis with a phase 1 vaccine (Coxevac, CEVA) had been shown to be very effective, especially when used before the first pregnancy (See Hogerwerf et al., 2011). Phase 2 vaccines should be discouraged. For Chlamydia abortus, the live attenuated vaccine has been described to be effective, but also abortion caused by the vaccine strain has been described. Nevertheless, is this vaccine preferable above the other available vaccine? In line 45-47 an advise around vaccination is given about C. burnetii and C. abortus. I would recommend to add somewhere in the manuscript the great differences between coxiellosis and enzootic abortion (transmission of pathogen, zoonotic risk, introduction in a flock, the presence of the bacterium, etc),
line 23 C. abortus has not been mentioned before and should be fully written here
line 23 I thought the name of the bacterium is Chlamydia abortus currently (please adapt in whole manuscript)
line 28 most instead of more
line 30 border disease virus should be written without capitols (please adapt in whole manuscript)
line 58 please add region where these are most common
line 72 dairy goats instead of small ruminants
line 91 Q fever instead of Q-fever
Line 92 notifiable in which species? Humans/animals??
line 123 1,242 ovine and 371 caprine cases and delete "sheep and goats respectively"
Why is not tested for Listeria spp?
Table 4 and 7 (0 and 1; I would rather see + and -). But the total look should be improved
Table 5 why here pestivirus in cluster 4? and not BDV?
line 266 a ref should be added
line 267-269 presence of a non-infectious cause because of abscence of positive PCR is not correct. These line should be changed! Only if you have done pathology and no inflammation was present, I can agree with you. Other possible abortifacient pathogens could have been missed!
line 279 hihly sensitive, but maybe to sensitive?
line 402 toxoplasmosis instead of Toxoplasmosis
line 404 salmonellosis instead of Salmonellosis
line 432-434 why should this be a bivalent vaccine?? The live attenuated should be administerd once in the three years and the phase 1 vaccine annually.... I miss here some practical considerations in this advise or do I miss something in your considerations?
Conclusions:
1. I would recommend some additional information on the importance of biosecurity in order to prevent introduction os pathogens via animals or other contact
2. I would recommend to advise to make a tailormade advise for countries/regions/farms based on their history and presence of certain pathogens.
Author Response
Reviewer 2.
Dear authors,
I enjoyed reading your manuscript. I do have some small suggestions. Two things that, to my opinion, should be discussed more extensively in the discussion are:
I do have some hesitations with the interpretation of Ct values of the Coxiella burnetii PCR. Low Ct values of this ubiquitous bacterium can also be caused by contamination (see Roest et al., 2012), especially in heavily contaminated environment this is a well know phenomenom. This ris for contamination is also the case for post mortem rooms. How sure are you that the samples with low Ct- values are not caused by contamination?
The question on whether the positive result in a sample corresponds to a contamination or to a real infection is almost impossible to solve, for these agents and for most of infectious agents involved in other pathologies. This concern affects most or all PCR diagnostics for infectious pathology. Moreover, some animals could carry a certain load of the agents while showing no symptoms at all. In the present work, samples are obtained from aborted fetuses or from females having suffered the abortion by veterinarians in the farms. Sterility, samples refrigeration, fast transport (always in 24h, with refrigeration) etc. are always our main concern and the samples are examined before being processed in order to detect any sign of degradation.
These questions were included in the discussion in the first version of the manuscript, lines 287-289: “...although definitive diagnosis should also consider other indicators such as histopathological lesions, microorganism cultures and specific staining techniques [28, 30]). As it refers to Ct threshold of 38, the value is decided in our laboratory according to the detection limits of the assay established in controlled conditions. In connection with this question, we have added the following paragraph:
“A cycle threshold (Ct) value of 38 was selected in the current study based on the limit of detection of the assay. It means that positive RT-qPCR results that are above to this threshold have a likelihood of being false positive, especially considering that different pathogens and specimens/tissues were analyzed. Another limitation of the study was related with the collection of specimens by veterinary clinics and the risk of contamination of samples in heavily contaminated environment which could even provide low Ct values in RT-qPCR. “
Some authors/institutions consider other limits: for instance, the IOE proposes for Coxiella a minimum charge of 10e4 units/mg of tissue, roughly equivalent to a Ct 33
- PCR detects a pathogen, but no causality with the abortion is shown since this manuscript does not contain pathological findings (and thus prove whether inflammatory signs fitting a certain pathogen is lacking).
As explained in the manuscript, this is a limitation in the work itself: the laboratory receives samples from different farms, but, in most cases examination of the females suffering the abortions is not available. The laboratory performs PCR diagnostics and identifies possible infectious agents involved in the abortions. As explained before, the question on the causality is always open, especially when detection relays on PCR. The limit for Ct values is based up on the detection limits of our systems, established in previous controlled assays. The following text, existing in the first version is related to this question:
“High qPCR values provide strong evidence that infectious abortions are caused by a specific pathogen and thus play a key role in the management of abortions in sheep and goat herds although definitive diagnosis should also consider other indicators such as histopathological lesions, microorganism cultures and specific staining techniques [28, 30].”
Another point is the advise to vaccinate against several pathogens. To my opinion more debate on the advantages and disadvantages of some of the vaccines should be added. Vaccination against coxiellosis with a phase 1 vaccine (Coxevac, CEVA) had been shown to be very effective, especially when used before the first pregnancy (See Hogerwerf et al., 2011). Phase 2 vaccines should be discouraged. For Chlamydia abortus, the live attenuated vaccine has been described to be effective, but also abortion caused by the vaccine strain has been described. Nevertheless, is this vaccine preferable above the other available vaccine? In line 45-47 an advise around vaccination is given about C. burnetii and C. abortus. I would recommend to add somewhere in the manuscript the great differences between coxiellosis and enzootic abortion (transmission of pathogen, zoonotic risk, introduction in a flock, the presence of the bacterium, etc),
Regarding this comment, we have made several modifications:
In the abstract, instead of :
“These findings indicate that both pathogens are the most significant ones to be combined in a bivalent vaccine to prevent abortions in small ruminants in this geographical area”
We now say:
“These findings indicate that both pathogens are the most significant ones to be readily prevented by vaccination in this geographical area. Biosecurity and biocontainment measures are also steady recommended to prevent both economic losses and public health risk associated with most of these abortifacient agents”
Besides, we have added the reference by Arricau-Bouvery et al., 2005 that you can see at number 92 now at the end of the discussion:
“In contrast, the phase II vaccine did not affect the course of the disease or excretion (92). The only authorized vaccine against…..”
line 23 C. abortus has not been mentioned before and should be fully written here
Thanks for your observation: we have written the full name in this point
line 23 I thought the name of the bacterium is Chlamydia abortus currently (please adapt in whole manuscript)
Thanks again for your observation. We have changed the name by Chlamydia abortus all over the manuscript.
line 28 most instead of more
Thanks again for your observation. The change is made
line 30 border disease virus should be written without capitols (please adapt in whole manuscript)
Thanks again for your observation. The change is made all over the manuscript.
line 58 please add region where these are most common
We have only accession to global information on this subject, not to regional data so that we can not make this change.
line 72 dairy goats instead of small ruminants
Thanks. The change is done.
line 91 Q fever instead of Q-fever
Thanks. The change is done.
Line 92 notifiable in which species? Humans/animals??
Thanks. The change is done.
line 123 1,242 ovine and 371 caprine cases and delete "sheep and goats respectively"
Thanks. The change is done.
Why is not tested for Listeria spp?
Listeria spp was included in the diagnostic panel for abortions some years ago. The prevalence was very low in our area, and as Listeria clinic signs are very characteristics, in these cases most practitioners directly demanded a single Listeria test instead of a full abortions panel diagnostics. For this reason, we finally decided to remove Listeria test from the panel.
Table 4 and 7 (0 and 1; I would rather see + and -). But the total look should be improved
Thanks . The 0 and 1 figures have been replaced by – and + at your request. Both tables 4 and 7 have been modified to provide more legibility.
Table 5 why here pestivirus in cluster 4? and not BDV?
The PCR assay used in this work is designed for pestivirus (both BVDV and BDV). Even in ovine samples, we can not affirm that positive PCR results are due only to BVD.
line 266 a ref should be added
In our copy, line 266 corresponds to a Table legend, and references are not usually added in such legends. However, from your next suggestions, we think you may refer to the lines 300 to 302 in our copy. In the line 302, the reference number 4 is presented. In spite of the difficulties concerning the changes in the number of the lines during the edition of the manuscript, we think that this and the next comment are related to the lines 300-313 in our copy, that is the first paragraph in the Discussion section. As you can see, the paragraph has been changed to avoid stating that our abortion samples negative for the seven tested agents are necessarily associated to non-infectious abortion, and several references are included:
The rate of abortion in healthy herds is usually below 2% but sheep and goat flocks can experience higher abortion rates sometimes resulting in “storms” which significantly affect productivity and require accurate diagnosis [4]. Some abortions in the present report may be induced by other possible abortifacient pathogens not included in our panel or by non-infectious causes, which have been reported to account for approximately 10% of events [43, 44]. Non-infectious causes were not investigated in this study. Among the 1,613 specimens studied in this work, a remarkable percentage tested negative to all the seven pathogens analyzed, namely more than 23% of cases from sheep and 33% from goats, suggesting a potential non-infectious aetiology or the action of other infectious agents. Several factors have been documented to cause abortions in livestock including toxic chemicals, genetic factors, metabolic and nutritional problems and physical factors. Non-infectious abortions are mostly sporadic and occur at any pregnancy stage, but causes are difficult to diagnose and in most cases remain unknown [43, 44]. Nevertheless, available evidence indicates that different bacteria, viruses and protozoa are by far the most plausible causative agents of abortions in sheep and goats [45].
.
line 267-269 presence of a non-infectious cause because of abscence of positive PCR is not correct. These line should be changed! Only if you have done pathology and no inflammation was present, I can agree with you. Other possible abortifacient pathogens could have been missed!
This is surely related to lines 300-311 in our copy, since the line numbers in your copy seems to be different to the ones in our copy. As stated before, the paragraph has been modified.
line 279 hihly sensitive, but maybe to sensitive?
The common term in the bibliography is “highly sensitive”. In our opinion, the term “too sensitive” is not appropriate for a diagnostics procedure.
line 402 toxoplasmosis instead of Toxoplasmosis
Thanks. The change is done.
line 404 salmonellosis instead of Salmonellosis
Thanks. The change is done.
line 432-434 why should this be a bivalent vaccine?? The live attenuated should be administerd once in the three years and the phase 1 vaccine annually.... I miss here some practical considerations in this advise or do I miss something in your considerations?
You are right: a bivalent vaccine should not be the best option in practice. For this reason, we have changed the lines 432-434.
Instead of “…… , therefore, should be the most appropriate to be combined in a bivalent vaccine to prevent abortions in small ruminants in this geographical area…”
we have written:
“ … therefore, should be the main targets of vaccination to prevent abortions in small ruminants in this geographical area. Additionally,…”
Conclusions:
- I would recommend some additional information on the importance of biosecurity in order to prevent introduction os pathogens via animals or other contact
We have added a phrase
“Biosecurity and biocontainment measures are also critical to prevent both economic losses and public health risk associated with most of these abortifacient agents”
at the conclusion section..
- I would recommend to advise to make a tailormade advise for countries/regions/farms based on their history and presence of certain pathogens.
We agree with you. We have modified the conclusions section to include this idea:
Moreover, the geographical variations in the ranking of infectious causes of abortion reported in other studies recommend investigations to establish the most common abortifacient agents to be prevented by vaccination at a local or regional level. Biosecurity and biocontainment measures are also critical to prevent both economic losses and public health risk associated with most of these infectious agents.

Reviewer 3 Report
This work is innovative and of great interest both scientifically and in terms of abortion management (diagnosis and consequently, targeting of prevention measures in small ruminant herds).
There is no equivalent work that includes so much quasi-systematic etiological research. However, some points need to be further detailed and explored.
The number of abortions studied was not related to the number of farms and the number of abortions investigated per abortive series. Consequently, it is impossible to take into account the weight of each farm and its etiological dominants in the results.
The research of pathogens could be performed on different tissues. No information was provided to indicate on which tissue each of these agents was tested. Indeed, the different pathogens have different tropisms and their dispersion in the tissues may also differ. A lack of evidence may not be synonymous with a lack of involvement in abortions if the matrix chosen is not relevant or is poorly collected (sampling area, quantity collected) or, subsequently, poorly preserved, poorly stored or with defects in the conditions of transport.
Furthermore, the choice of a Ct threshold of 38 can be questioned at least for some pathogens and depending on the tissues studied. For example, the existence of high vaginal excretion of Chlamydia or Coxiella in animals that have not aborted should prompt consideration of pathogen quantification, which has not been done. This raises the question of the actual frequency of coinfections in the sense that there may have been a dominant pathogen causing abortion concurrently with persistent shedding of another pathogen. This could at least be discussed.
The time between the abortions and the taking of samples and subsequent analysis was not indicated. However, a strong decrease in bacterial loads has been demonstrated in some tissues (especially in vaginal swabs) beyond 48 hours after abortion.
The distinction between individual cases and abortive episodes, the distinction between co-infection and co-circulation of pathogens, and the description of the number and nature of samples per abortion (maternal or fetal samples) would need to be clarified to clarify the overall diagnostic procedure.
Finally, it is unfortunate not to consider the interest of indirect diagnostic analyses (search for evidence of recent circulation of pathogens on a herd scale in certain groups of individuals, seroconversion or evolution of antibody titres) to complete the direct diagnosis and possibly consolidate the results obtained.
There is no doubt that this type of study is extremely valuable and for this reason it is essential to document the approach adopted in a more complete way. The conclusions reached should also be reviewed in the light of the limitations of the approach (false negatives, false positives, limitations of the analyses themselves or of the realization and management of the samples).
Author Response
Reviewer 3
This work is innovative and of great interest both scientifically and in terms of abortion management (diagnosis and consequently, targeting of prevention measures in small ruminant herds).
There is no equivalent work that includes so much quasi-systematic etiological research. However, some points need to be further detailed and explored.
The number of abortions studied was not related to the number of farms and the number of abortions investigated per abortive series. Consequently, it is impossible to take into account the weight of each farm and its etiological dominants in the results.
The work is based on test performed on demand. For this reason, the identity of the herd is rarely disclosed by the veterinarians submitting the samples. However, when data on this question are available, two to five samples are received from each single farm.
The research of pathogens could be performed on different tissues. No information was provided to indicate on which tissue each of these agents was tested. Indeed, the different pathogens have different tropisms and their dispersion in the tissues may also differ. A lack of evidence may not be synonymous with a lack of involvement in abortions if the matrix chosen is not relevant or is poorly collected (sampling area, quantity collected) or, subsequently, poorly preserved, poorly stored or with defects in the conditions of transport.
Different tropisms of the different agents are taken into account. For this reason, when fetuses are sent to the laboratory, a mix of placent, encephalon, lung, liver and stomach content is prepared in order to extract nucleic acids: in this way, we try to include the maximum number of susceptible target tissues. Precise instructions are given to practitioners in order to refrigerate samples, which are sent in isolated boxes together with refrigerating frozen blocks and are expected to reach our laboratory in one day, a routine service for most transport companies nowadays in our area of study. The general condition of the samples is always evaluated so that those eventually showing signs of degradation should be excluded from the analysis. As it refers to vaginal swabs, they were always taken from females having suffered abortions in the last 48-72 hours as a maximum. Subsequent transport of the swabs is subjected to the same procedure. These ideas have now been included in the Material and methods section.
However, the authors could not oblige the different practitioners all over the Iberian Peninsula to send a particular kind of sample, even if sampling follows the traditional professional procedures for abortion diagnostics. After an internal discussion, the authors have decided not to include data on this subject in the manuscript, to prevent any bias or mistake
Furthermore, the choice of a Ct threshold of 38 can be questioned at least for some pathogens and depending on the tissues studied. For example, the existence of high vaginal excretion of Chlamydia or Coxiella in animals that have not aborted should prompt consideration of pathogen quantification, which has not been done. This raises the question of the actual frequency of coinfections in the sense that there may have been a dominant pathogen causing abortion concurrently with persistent shedding of another pathogen. This could at least be discussed.
The question on whether the positive result in a sample corresponds to a contamination or to a real infection is almost impossible to solve in practice, for these agents and for most of infectious agents involved in other pathologies. Moreover, some animals could carry a certain load of the agents while showing no symptoms at all. The question of dominant pathogens causing shedding of other pathogens is not limited to PCR procedures and should require additional experiments not affordable by either farmers or by the authors. These questions were included in the discussion in the first version of the manuscript, lines 287-289: “...although definitive diagnosis should also consider other indicators such as histopathological lesions, microorganism cultures and specific staining techniques [28, 30]) .
As it refers to Ct threshold of 38, the value is decided according to the detection limits of the assay determined for our procedure. For this reason, we have included the phrase:
“A cycle threshold (Ct) value of 38 was selected in the current study based on the limit of detection of the assay. It means that positive RT-qPCR results that are above to this threshold have a likelihood of being false positive, especially considering that different pathogens and specimens/tissues were analyzed. Another limitation of the study was related with the collection of specimens by veterinary clinics and the risk of contamination of samples in heavily contaminated environment which could even provide low Ct values in RT-qPCR. “
Some authors/institutions consider other limits: for instance, the IOE proposes for Coxiella a minimum charge of 10e4 units/mg of tissue, roughly equivalent to a Ct 33.
The time between the abortions and the taking of samples and subsequent analysis was not indicated. However, a strong decrease in bacterial loads has been demonstrated in some tissues (especially in vaginal swabs) beyond 48 hours after abortion.
As stated before, precise instructions are given to practitioners in order to refrigerate samples, which are sent in isolated boxes together with refrigerating frozen blocks and are expected to reach our laboratory in one day, a routine service for most transport companies nowadays in our area of study. The general condition of the samples is always evaluated so that those eventually showing signs of degradation should be excluded from the analysis. We are adding a phrase to the material and methods section:
“Samples are immediately refrigerated and delivered to the laboratory via urgent transport in 24hours. Vaginal swabs were obtained within the 48h following abortion and urgently transported too.”
The distinction between individual cases and abortive episodes, the distinction between co-infection and co-circulation of pathogens, and the description of the number and nature of samples per abortion (maternal or fetal samples) would need to be clarified to clarify the overall diagnostic procedure.
This is one of the limitations of our study, since the diagnostics is performed on demand and, in many cases, precise data on the origin of the samples is not available. However, according to our clinical experience, individual abortions in ovine and caprine are given little or no attention at all. We can therefore accept that most cases proceed from abortive episodes. However, since this is not confirmed in most cases, we have preferred not making any direct comment on it. As it refers to the kind of samples, many submissions combine several different samples: fetus+ placenta, fetus+ vaginal swabs etc…In all, we have studied 830 fetuses, 582 vaginal swabs, 235 placentas, six isolated organs and 395 sera (these last ones always accompanied by other samples). In each case, the laboratory provides the herd manager with a diagnostics, and this is the one we have used in our analysis. Specifying the kind of sample received in each case and trying to obtain clear conclusions related to it was at present impossible, and we have preferred a simplification giving place to a clear idea of the situation.
Finally, it is unfortunate not to consider the interest of indirect diagnostic analyses (search for evidence of recent circulation of pathogens on a herd scale in certain groups of individuals, seroconversion or evolution of antibody titres) to complete the direct diagnosis and possibly consolidate the results obtained.
We agree that a complete story for each case, involving seroconversion, antibody titers etc should have provided more detailed information on this disease. However, in the last years, routine diagnosis is based up on PCR for economy and efficiency reasons. Ovine and caprine farming are low income activities and owners can’t usually afford this kind of global control. The authors did not receive such data and, visiting the different farms to obtain this information was also impossible due to the geographical distances. In different sections of the manuscript the convenience of other diagnostic procedures is clearly reflected.
There is no doubt that this type of study is extremely valuable and for this reason it is essential to document the approach adopted in a more complete way. The conclusions reached should also be reviewed in the light of the limitations of the approach (false negatives, false positives, limitations of the analyses themselves or of the realization and management of the samples).
We have modified the conclusions in the light of your comments. Also, the material and methods section reflects some of the limitations of the assay.
